# Correlations of Accelerometer-Measured Physical Activity with Body Image and Quality of Life among Young and Older Adults: A Pilot Study

**DOI:** 10.3390/ijerph192214970

**Published:** 2022-11-14

**Authors:** Amir Dana, Sheida Ranjbari, Hasan Mosazadeh, Wojciech Jan Maliszewski, Aleksandra Błachnio

**Affiliations:** 1Department of Physical Education and Sport Sciences, Tabriz Branch, Islamic Azad University, Tabriz 5157944533, Iran; 2Department of Physical Education, Urmia Branch, Islamic Azad University, Urmia 5716963896, Iran; 3Department of Psychology, Kazimierz Wielki University, 85-064 Bydgoszcz, Poland; 4Faculty of Social Work, University of Applied Sciences, 64-920 Piła, Poland

**Keywords:** adult, physical activity, body image, quality of life, aging

## Abstract

Significant evidence suggests that regular physical activity (PA) is correlated with numerous psychological benefits in adults such as improving body image and quality of life. However, this issue has not been differentiated between young and older adults. In addition, most previous studies used a self-reported questionnaire for measuring PA, the objectivity of which is limited in several ways. Hence, by using accelerometer technology for monitoring PA, this study was designed to examine the correlations of PA with body image and quality of life in young and older adults. In this cross-sectional study, we used objective actigraphy and survey data from 147 young and older adults, including 77 young and 70 older adults from Tehran, Iran. To examine our variables and hypothesis, the following instruments were implemented: the Persian version of the Multidimensional Body-Self Relations Questionnaire (MBSRQ), the Persian version of the Quality-of-Life Questionnaire (WHOQOL-BREF), and the ActiGraph wGT3X-BT for measuring PA. An independent t-test and a multivariate regression analysis were used to analyze the data. The weekly PA of both young and older adults was found to be lower than the recommended amount. Young adults engaged in significantly more weekly PA than older adults. For young adults, PA (including MPA, VPA, and MVPA) was generally found to be significantly correlated with body image and quality of life. For older adults, however, we found only significant correlations between VPA and quality of life. These findings indicated that PA is a critical concern in adults, particularly older adults. Accordingly, it is necessary to adopt appropriate strategies to promote an active lifestyle among adults.

## 1. Introduction

Physical activity (PA) is defined as any movement of the body produced by skeletal muscles that leads to energy expenditure [1,2]. Significant evidence suggests that regular PA is correlated with numerous health benefits such as improved cardiorespiratory and muscular fitness, strong bones, enhanced memory function and cognitive control, and reductions in depressive symptoms and obesity [3,4,5,6]. Therefore, World Health Organization (WHO) guidelines and recommendations provide details for different age groups and specific populations regarding the amount of PA needed for good health. In the case of adults, the WHO recommends that adults (aged 18–64 years) should engage in at least 150–300 min of moderate-intensity PA (MPA) or at least 75–150 min of vigorous-intensity PA (VPA) or 150 min of moderate-to-vigorous-intensity PA (MVPA) per week [7]. Nevertheless, evidence indicates that very few adults meet the WHO guidelines and that most of them spend the majority of time as sedentary [8,9]. For example, Du et al. [8] found that 65.2% of American adults met the PA guidelines for at least 150 min a week of MPA or 75 min a week of VPA or an equivalent combination of MVPA. In addition, Loye et al. [9] found that European adults from England, Portugal, Norway, and Sweden were sedentary for 530 min/day and accumulated 36 min/day of MVPA. A total of 23% accumulated more than 10 h of sedentary time/day, and 72% did not meet the physical activity recommendations. Due to the low level of PA in adults, international data revealed that the global prevalence of overweight and obesity increased in adults between 1990 and 2017 [10], which may consequently increase the experience of dissatisfaction with physical appearance [11,12]. As is obvious, previous studies have clearly shown low levels of PA among adults; however, there is a paucity of data on activity in young and old adults across genders. For example, it has been shown that during young adulthood (ages 20–30), total and light-intensity PA increases with age and then stabilizes during midlife (ages 31–59) [13]. The primary purpose of the present study was to further examine this issue and to monitor the PA levels among young and old adults.

PA is one of the various factors that have a positive effect on body image (BI). BI is a multidimensional construct that refers to an individual’s subjective representation of their body [14,15,16]. Evidence showed that a negative BI may be correlated with a high level of mental disorders such as anxiety and depression, as well as low self-esteem [17,18]. Moreover, self-reported studies demonstrated that participating in PA can positively affect perceived BI among adults [11,16]. There are two points that warrant the need for further research. First, previous studies have mostly used self-reported questionnaires to measure PA, which results in a self-reporting bias. In fact, it was shown that there was a significant difference between the data obtained from a questionnaire and from the accelerometer device [19] that was used to objectively measure the PA, and that the questionnaire was not able to accurately measure MVPA. Therefore, to show the correlations between PA and BI more accurately, it is necessary to use an accelerometer to measure PA. In addition, in previous studies the correlations between PA and BI were investigated in adults with an age range of early ages (i.e., young adults, 18–35 years) to late ages (i.e., older adults, older than 65 years). It was reported that when adults experience several health-related problems such as cancer, hypertension, osteoporosis, stroke, diabetes, muscle and skeletal pain, etc. [4,5], it can lead to feeling a gradual shift in concerns from physical appearance to health and bodily functioning. This makes it necessary to examine the correlations between PA and BI in young and older adults separately. Hence, the second purpose of this study was to examine the correlations between accelerometer-measured PA with BI among young and older adults.

In addition, it was shown that PA is a strong predictor of quality of life (QoL) [20]. QoL is a broad multidimensional concept that refers to an individual’s perception of their position in life and opportunities for happy and active participation in sociocultural, economic, and political life [21,22]. Improving the quality of life is widely regarded as a priority area of health interventions even if the economic status and social conditions of countries are not conducive to such actions [23]. Specifically, QoL focuses on the subjective self-perception of current health status and the ability to perform daily activities in different life domains [24,25]. Evidence from systematic reviews indicated that higher levels of PA were correlated with higher QoL scores in healthy populations, including adults [11,20,25,26]. However, similar to BI, this correlation was not based on accelerometer-measured PA and did not differentiate between young and older adults. Hence, the third purpose of the present study was to examine the correlations between accelerometer-measured PA and QoL among young and older adults.

In total, this study aimed: (1) to compare the accelerometer-measured PA pattern, BI, and QoL between young and older adults; and (2) to examine the correlations of accelerometer-measured PA with BI and QoL among young and older adults. We hypothesized that young adults would have a higher weekly PA, BI, and QoL than older adults. Moreover, we hypothesized that PA would correlate with BI and QoL in young and old adults.

## 2. Materials and Methods

### 2.1. Participants

A total of 147 Iranian adults including 77 young adults (37 women) aged 22 to 35 years with an average age of 27.61 years (SD = 5.20) and 70 older adults (32 women) aged 60 to 69 years with an average age of 64.57 years (SD = 3.18) voluntarily participated in this study. The participants lived in Tehran, the capital of Iran. A representative sample of young and older adults was recruited in the same manner, which included word of mouth and advertisements to the general public on social networks such as Telegram and WhatsApp. One accepting via phone contact, the self-administered written consents were distributed to the interested participants and collected after completion. Participation was voluntary and anonymous, and by completing the written consents, the participants agreed to participate. The inclusion criteria were being at least 18 years old as well as being physically and mentally healthy without any chronic diseases. The exclusion criteria included having any acute physical or mental illnesses. In addition, we excluded those who severely experienced COVID-19. Participants spoke Persian as their mother language. The sample-size calculation was based on previous cross-sectional studies that used accelerometers for measuring PA in adults [27,28]. The initial sample of this study was 207 adults; however, 60 participants were excluded from the study due to not completing the accelerometer procedures (Figure 1). The study was conducted in accordance with the declaration of Helsinki, and the University Ethics Committee approved the research protocol (Code: IR.IAU.AK.REC.1400.001). Participants were informed about all study procedures and gave written informed consent.

### 2.2. Measures

#### 2.2.1. Physical Activity

PA was measured objectively using an ActiGraph wGT3X-BT accelerometer (ActiGraph LLC, Pensacola, FL, USA). Accelerometers measure the frequency, intensity, and duration of PA, as well as the sedentary time. The ActiGraph accelerometer has good validity and reliability [29,30]. The participants wore an accelerometer for seven consecutive days and then the data were downloaded, processed, and analyzed using the software ActiLife v6.13.4 (Actigraph Inc., Pensacola, FL, USA). The mean values at each PA intensity were calculated using the cutoff points proposed by Freedson et al. [31]: light PA (100–1951 counts/min), MPA (≥1952–5724 counts/min), and VPA (≥5725 counts/min). MPA and VPA were merged into MVPA.

#### 2.2.2. Body Image

The Persian version of short form of Multidimensional Body-Self Relations Questionnaire-Appearance Scales (MBSRQ-AS) [32,33] was used for evaluating BI. MBSRQ-AS is a self-reported inventory that assesses peoples’ attitudes toward the different aspects of body image and is intended to be used by adults. The short form of the MBSRQ-AS is a 34-item measure that consists of five subscales, namely Appearance Evaluation (7 items), Appearance Orientation (12 items), Overweight Preoccupation (4 items), Self-Classified Weight (2 items), and the Body Areas Satisfaction Scale (9 items). Each item was scored on a 5-point scale and evaluates agreement (from 1: “definitely disagree” to 5: “definitely agree”), frequency (from 1: “never” to 5: “very often”), or satisfaction (from 1: “very dissatisfied” to 5: “very satisfied”). For items related to Self-Classified Weight, participants use ratings from 1: “very underweight” to 5: “very overweight”. High scores on this measure indicated a higher satisfaction with the general body image. Internal consistencies of the subscales of the original scale ranged from 0.76 to 0.86 [32,33]. The reliability of the Persian version of the MBSRQ-AS was also confirmed by a Cronbach’s alpha of the total scale of 0.83 [34] and 0.98 [35]. In this study, the Cronbach’s alpha coefficient of the total scale was 0.92.

#### 2.2.3. Quality of Life

The Persian version of the WHOQOL-BREF questionnaire was used to assess the QoL [36]. It consisted of 26 questions, of which 24 were divided in four domains: physical health, psychological health, social relationships, and environment; the remaining 2 questions measured the self-perceived QoL and satisfaction with health. Each domain was represented by several facets and questions were formulated for a Likert response scale with intensity (nothing—extremely), capacity (nothing—completely), frequency (never—always), and assessment scales (very dissatisfied—very satisfied; very bad—very good), all of which consisted of five levels (one to five). The obtained raw score was transformed on a scale from 0 to 100 to enable comparisons to be made between domains [36]. A reliability of this scale of 0.75 to 0.84 was obtained in four categories [36]. The reliability of the Persian version was as follows: Physical domain = 0.77, mental/psychological domain = 0.77, range of social relations = 0.75, and environmental health = 0.84 [37]. In another study, the reliability of the Persian version of the WHOQOL-BREF was also confirmed by a Cronbach’s alpha of the total score of 0.70 [38]. In this study, the Cronbach’s alpha coefficient of the total score of this scale was 0.88.

### 2.3. Data Analysis

We analyzed the data by using SPSS Statistics version 26 (SPSS Inc., Chicago, Ill., USA). A descriptive analysis was used to calculate the means and standard deviations of the PA pattern, BI, and QoL. A chi-squared test was used to compare demographic data including financial status, education, and employment between young and older adults. An independent t-test was used to compare research variables among young and old adults. A multivariate regression analysis was utilized to measure the bidirectional correlations between the research variables. Here, we considered financial status and education as covariates. Significant levels were considered at an alpha level of 0.05.

## 3. Results

### 3.1. Demographic Data

Table 1 shows the demographic characteristics of the study’s sample. As is obvious, most of young and older adults were at a medium level of financial status. Here, the results of the chi-squared test showed no significant differences between young and older adults (*p* > 0.05). Young adults mostly had a college education. However, about half of the older adults had a college education. Here, the results of the chi-squared test showed that the young adults had a significantly higher educational level than the older adults (*p* = 0.000). Finally, most of the young adults were employed, while most of the older adults were retired. Here, the results of the chi-squared test showed no significant differences between young and older adults (*p* > 0.05).

### 3.2. Young vs. Old Differences

Table 2 shows the means and standard deviations of the PA pattern, BI, and QoL among young and older adults. The accelerometer data showed that compared with older adults, young adults had a significantly higher light PA (283.12 vs. 142.76 min/week for young and older adults, respectively; *p* < 0.001), MPA (95.45 vs. 68.83 min/week for young and older adults, respectively; *p* = 0.003), VPA (33.13 vs. 18.01 min/week for young and older adults, respectively; *p* < 0.001), and MVPA (128.59 vs. 86.84 min/week for young and older adults, respectively; *p* < 0.001). In addition, the data showed that compared with older adults, young adults had significantly higher scores for BI and its subscales, which included Appearance Evaluation, Appearance Orientation, Overweight Preoccupation, Self-Classified Weight, and the Body Areas Satisfaction Scale (all *p* < 0.05). Finally, our findings showed that compared with older adults, young adults had significantly higher scores for QoL and some subscales including physical health, psychological health, and environment (all *p* < 0.05). 

### 3.3. Bidirectional Correlations

The results of the multivariate regression analysis are shown in Table 3 and Table 4 for young and older adults, respectively. Our findings showed that for young adults, generally PA (including MPA, VPA, and MVPA) was significantly correlated with BI and QoL (all *p* < 0.05). For older adults, however, we found only significant correlations between VPA and QoL (all *p* < 0.05).

## 4. Discussion

This study was, to the best of our knowledge, the first to examine the correlations of accelerometer-measured PA with BI and QoL among young and older adults. Regarding the PA pattern, the accelerometer data showed that compared with older adults, young adults had significantly higher light PA (283.12 vs. 142.76 min/week for young and older adults, respectively), MPA (95.45 vs. 68.83 min/week for young and older adults, respectively), VPA (33.13 vs. 18.01 min/week for young and older adults, respectively), and MVPA (128.59 vs. 86.84 min/week for young and older adults, respectively). These findings confirmed our first hypothesis. As mentioned earlier, there was no direct prior knowledge of a comparison of the PA patterns of young and older adults; however, some evidence showed that the prevalence of specific higher-intensity physical activities decreased with age among adults, while the prevalence of reported inactivity showed an age-related increase that was especially evident among women [39,40]. The lower participation in PA of older adults compared to younger adults is probably due to older adults’ lower physical abilities (e.g., a lesser degree of physical fitness, lower functional abilities, lower muscular and cardiovascular abilities, etc.). In addition, compared with the WHO guidelines, our findings showed that young adults did not meet the guidelines of at least 150–300 min of MPA or 75–150 min of VPA or 150 min of MVPA per week. These findings were in line with previous studies worldwide [41,42,43,44,45] that showed that young adults did not follow the recommended amount of PA per week. Similar findings were observed for older adults, indicating that older adults also did not follow international guidelines for PA.

Regarding BI, the data showed that compared with older adults, young adults had significantly higher scores for BI and its subscales, which included Appearance Evaluation, Appearance Orientation, Overweight Preoccupation, Self-Classified Weight, and the Body Areas Satisfaction Scale. These results were in line with those of previous studies [46,47,48,49,50]. Among the reasons for these findings, it can be stated that older adults had less anxiety regarding their appearance and had a lower drive for thinness and less restricted eating. Older adults placed less importance on their body, leading to less self-objectification in comparison to younger adults [46,49,50]. Finally, we found that in young adults, PA (e.g., MPA, VPA, and MVPA) was correlated with BI and all its subscales. However, in older adults, we did not generally observe significant correlations between PA and BI. This was one of the interesting results of the current study because it showed the different effects of PA on BI in young and older adults. In addition, these results could indicate that PA in young adults is an influencing factor on people’s perception of their body image, but in older adults, this is not the case. More studies are needed to determine the effects of PA on BI in young and older adults. The mechanisms behind the effects of PA on BI are not well understood; however, evidence suggested that PA may result in actual changes in one’s body shape and/or weight, perceived changes in one’s shape and/or weight, and improved perceptions of self-efficacy [15,16].

Concerning QoL, the data showed that compared with older adults, young adults had significantly higher scores for QoL and some subscales including physical health and environment. Living alone, missing important people in life, lower physical and mental health, and less social and emotional support may be among the possible reasons for a lower QoL among older adults than younger adults [22,24,51]. Finally, we found that in young adults, PA (e.g., MPA, VPA, and MVPA) was correlated with QoL and all its subscales. In older adults, we observed a significant correlation between VPA and QoL. These findings confirmed our hypothesis and were in line with the findings of previous studies [20,26,29] indicating that PA had a positive impact on QoL among adults. This was quite understandable due to the fact that PA has numerous physical, psychological, and mental benefits [3,4,5,6]. Hence, people with a higher health status perceived higher levels of QoL [20,26,29]. On the other hand, although several factors affect a person’s perception of their QoL, at least based on the results of the present study, it can be stated that lower PA levels of older adults in comparison to young adults may explain the lower QoL of older adults.

The strengths of our study were: first, the use of up-to-date accelerometers to objectively determine the amount and levels of PA of young and older adults, which made it possible to prevent typical biases that are often correlated with self-reporting methods; secondly, we measured PA, BI, and QoL in both young and older adults. However, with 147 participants, the sample size should be considered as critically small. In addition, due to the use of questionnaires to measure BI and QoL, it should be noted that the questionnaires had a self-reporting bias limitation.

## 5. Conclusions

PA and psychological components such as BI and QoL have rarely been compared in young and older adults. In this study, we attempted for the first time to examine the correlations of accelerometer-measured PA with BI and QoL among young and older adults. Our findings showed that young and older adults did not meet the recommended amount of PA, which makes it necessary to adopt strategies to enhance the level of PA in adults. In addition, young adults were more physically active than older adults. These findings indicated that strategies to enhance PA should have a special focus on older adults. Moreover, young adults perceived a higher BI and QoL than older adults. Strengthening the factors affecting BI and QoL is essential, especially in older adults. Finally, PA was positively correlated with both BI and QoL, indicating that enhancing the level of PA among adults is necessary. Altogether, these findings indicated that PA is a critical concern for adults, particularly for older adults. Our findings have practical implications as well. For example, health education interventions and programs to promote an active lifestyle should be encouraged among adults. Although the current research was conducted on adults in Iran, these results can be used in other countries as well. However, in order to generalize these results to other countries, cultural and social differences, as well as social welfare facilities in different societies, should also be considered. Finally, it should be noted that this study was limited in the sense of our relatively small sample size.

## Figures and Tables

**Figure 1 ijerph-19-14970-f001:**
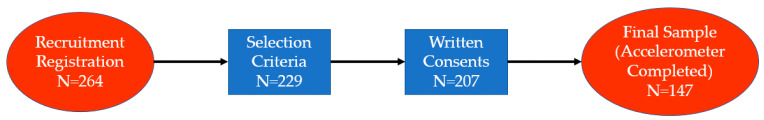
Flow diagram of the sample recruitment.

**Table 1 ijerph-19-14970-t001:** Demographic characteristics of the study’s sample.

Variables	Young Adults (n = 77)	Older Adults (n = 70)
Age (years)	27.91 ± 5.42	64.84 ± 3.02
Financial status		
Low	14(18%)	8(11%)
Medium	57(74%)	56(80%)
High	6(8%)	6(9%)
Education		
High school and below	7(9%)	34(49%)
College	70(91%)	36(51%)
Employment		
Unemployed	6(8%)	11(16%)
Employed	71(92%)	27(38%)
Retired	-	32(46%)

**Table 2 ijerph-19-14970-t002:** Comparison of PA pattern, BI, and QoL among young and older adults.

Variables		Groups	Mean	SD	Age Differences
Light PA (min/week)		Young (n = 77)	283.12	112.51	t = 7.35 *p* < 0.001 ***
		Old (n = 70)	142.76	118.97	
Moderate PA (min/week)		Young (n = 77)	95.45	55.41	t = 3.05 *p* = 0.003 **
		Old (n = 70)	68.83	49.62	
Vigorous PA (min/week)		Young (n = 77)	33.13	23.02	t = 4.42 *p* < 0.001 ***
		Old (n = 70)	18.01	17.77	
MVPA (min/week)		Young (n = 77)	128.59	69.32	t = 6.01 *p* < 0.001 ***
		Old (n = 70)	86.84	56.23	
Body Image	Appearance Evaluation	Young (n = 77)	3.99	0.84	t = 3.00 *p* = 0.003 **
		Old (n = 70)	3.51	1.09	
	Appearance Orientation	Young (n = 77)	3.38	0.98	t = −2.548 *p* = 0.013 *
		Old (n = 70)	2.83	1.20	
	Overweight Preoccupation	Young (n = 77)	3.14	1.03	t = 2.93 *p* = 0.004 **
		Old (n = 70)	2.58	1.26	
	Self-Classified Weight	Young (n = 77)	2.94	0.96	t = 2.93 *p* = 0.004 **
		Old (n = 70)	2.42	1.15	
	Body Areas Satisfaction Scale	Young (n = 77)	2.98	1.03	t = 2.87 *p* = 0.005 **
		Old (n = 70)	2.44	1.22	
	Total Score	Young (n = 77)	3.29	0.88	t = 3.22 *p* = 0.002 **
		Old (n = 70)	2.76	1.09	
Quality of Life	Overall QoL	Young (n = 77)	58.42	17.09	t = 1.55 *p* = 0.123
		Old (n = 70)	53.79	19.10	
	General Health	Young (n = 77)	56.16	16.83	t = 1.88 *p* = 0.061
		Old (n = 70)	50.67	18.48	
	Physical Health	Young (n = 77)	55.17	16.50	t = 2.78 *p* = 0.006 **
		Old (n = 70)	47.59	16.46	
	Psychological Health	Young (n = 77)	53.33	16.45	t = 2.79 *p* = 0.006 **
		Old (n = 70)	45.84	16.00	
	Social Relationships	Young (n = 77)	56.79	16.30	t = 1.74 *p* = 0.083
		Old (n = 70)	51.92	17.52	
	Environment	Young (n = 77)	48.18	16.62	t = 2.47 *p* = 0.015 *
		Old (n = 70)	41.62	15.48	
	Total Score	Young (n = 77)	54.68	14.16	t = 2.27 *p* = 0.025 *
		Old (n = 70)	48.57	16.42	

* *p* < 0.05; ** *p* < 0.01; *** *p* < 0.001.

**Table 3 ijerph-19-14970-t003:** Results of correlations between variables for young adults with financial status and education as covariates.

Variable	BI			QoL		
	Adjusted OR	95% CI	*p*	Adjusted OR	95% CI	*p*
Light PA	0.96	1.15–1.35	0.114	0.85	1.07–1.28	0.239
MPA	1.61	1.62–2.15	<0.001	1.59	1.56–2.07	<0.001
VPA	1.47	1.29–1.86	0.008	1.39	1.34–1.98	0.004
MVPA	1.75	1.86–2.58	<0.001	1.87	1.75–2.41	<0.001

BI: body image; QoL: quality of life; OR: odds ratio; CI: confidence interval.

**Table 4 ijerph-19-14970-t004:** Results of correlations between variables for older adults with financial status and education as covariates.

Variable	BI			QoL		
	Adjusted OR	95% CI	*p*	Adjusted OR	95% CI	*p*
Light PA	0.47	1.26–1.53	0.368	0.51	1.22–1.43	0.348
MPA	0.61	1.35–1.53	0.330	0.72	1.55–1.90	0.294
VPA	0.69	1.57–1.80	0.302	1.78	1.94–2.63	<0.001
MVPA	0.77	1.38–1.84	0.287	0.55	1.45–1.71	0.340

BI: body image; QoL: quality of life; OR: odds ratio; CI: confidence interval.

## Data Availability

The data presented in this study are not publicly available due to ethical considerations.

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
