# Peer review of "Correlations of Accelerometer-Measured Physical Activity with Body Image and Quality of Life among Young and Older Adults: A Pilot Study"

_ijerph, 2022, doi:10.3390/ijerph192214970_

Round 1
Reviewer 1 Report (Previous Reviewer 3)
Thank you for the opportunity to re-review the manuscript „Associations between Accelerometer Measured Physical Activity with Body Image and Quality of Life among Young and Older Adults“. This study aimed to to examine the associations between physical activity with body image and quality of life in young and older adults.
I have major concerns:
The authors should add the recruitment process of the study participants. The authors should add the inclusion and exclusion criteria of this study. The authors should add real methods for calculating the representative sample size. According to the recruitment process of this study, the authors should add a flow diagram of this study.
„Associations“ must be changed to „correlations“. Did the authors perform multivariate analysis?
The authors reported relatively small sample size as a main limitation. If it is not possible to generalize the study results, I suggest that the authors consider a pilot study.
Kind Regards
Author Response
Response to Reviewer 1 Comments
Point 1: The authors should add the recruitment process of the study participants. The authors should add the inclusion and exclusion criteria of this study. The authors should add real methods for calculating the representative sample size. According to the recruitment process of this study, the authors should add a flow diagram of this study.
Response 1: First of all, thank you very much for re-reviewing this paper.
The proposal has been followed. Recruitment process is clearly explained. As well, inclusion and exclusion criteria were added. A flaw diagram for study sample was added. Concerning sample size formula, we should be honest and say that we did not use any calculation formula for sample size. We tried to keep the sample size of our study consistent with previous research. Relevant references were added in the text. Please see Method.
Point 2: „Associations“ must be changed to „correlations“. Did the authors perform multivariate analysis?
Response 2: It has been corrected in whole the text. Associations were changed to correlations. In addition, we used multivariate regression analysis to measure bidirectional correlations between variables with considering financial status and education as covariates.
Point 3: The authors reported relatively small sample size as a main limitation. If it is not possible to generalize the study results, I suggest that the authors consider a pilot study.
Response 3: The proposal has been followed. Please see Title.
Reviewer 2 Report (New Reviewer)
The study entitled “Associations between Accelerometer-Measured Physical Activity with Body-Image and Quality of Life among Young and Older Adults”. It Is a well-structured manuscript with novelty data research. But several issues should be addressed.
1. Line 77, why did the authors define older adults as older than 55 years? It commonly uses 65 years as older adults. Please provide strong evidence to support your opinion.
2. In statistical analysis, I am concerned that the associations between each variable were examined, why did not the author consider the covariates (e.g., financial status, education) that may influence the bidirectional relationship? Please clarify it.
3. Some limitations should be addressed in your study. for example, how did you ensure the accuracy of self-reported measurements? Can the research results accurately reflect the direct relationship between variables?....... please address it carefully.
4. How are the results applied in practice? Please make a statement/recommendation based on your results.
Author Response
Response to Reviewer 2 Comments
Point 1: Line 77, why did the authors define older adults as older than 55 years? It commonly uses 65 years as older adults. Please provide strong evidence to support your opinion.
Response 1: First of all, thank you very much for reviewing our paper.
It has been corrected in the text. Please see Introduction.
Point 2: In statistical analysis, I am concerned that the associations between each variable were examined, why did not the author consider the covariates (e.g., financial status, education) that may influence the bidirectional relationship? Please clarify it.
Response 2: Proposal has been followed. We used multivariate regression analysis to measure bidirectional correlations between variables with considering financial status and education as covariates.
Point 3: Some limitations should be addressed in your study. for example, how did you ensure the accuracy of self-reported measurements? Can the research results accurately reflect the direct relationship between variables?....... please address it carefully.
Response 3: We added accuracy of self-reported measurements as a limitation of the study. Please see Discussion.
Point 4: How are the results applied in practice? Please make a statement/recommendation based on your results.
Response 4: We added practical implications of our results into the text. Please see Conclusion.
Round 2
Reviewer 1 Report (Previous Reviewer 3)
The Authors answered my questions.
Kind Regards
Reviewer 2 Report (New Reviewer)
All comments have been modified, and the modification is relatively perfect.
This manuscript is a resubmission of an earlier submission. The following is a list of the peer review reports and author responses from that submission.
Round 1
Reviewer 1 Report
In this study, gender and age differences in physical activity (PA), body image (BI) and QoL are investigated and, for the total sample, the interrelations are examined. The authors used accelerometer data to measure physical activity. In the introduction, the authors define the outcome variables well, but the need and novelty of the study remain unclear and the rationale and research question fit insufficiently. The method is well structured, but is missing information. Sample size is relatively low. The results are only tested on a basic level. I have the following specific concerns:
1. General: English language should be checked carefully for complete manuscript
2. Introduction line 48-60. Apparently, the major point is that the relation between PA and BI may depend on age. However, this is not examined in this paper. In the paper only differences in means are tested for age.
3. Introduction line 53-56: However, differences in the associations between PA and BI …not fully understood. What exactly is not understood? What information is missing? How does this affect knowledge? Please provide references as well.
4. Introduction line 56-61: This point can be elaborated more extensively. Differences in mean scores of self-report vs. accelerometry is not the main problem, but the relationship. How is the relationship? How may this affect relationships with BI and QoL?
5. Introduction line 71-72: similar to point 2., age differences in relationships are not tested.
6. Introduction: The authors do not give a rationale to examine gender differences.
7. Introduction: The research questions do not fit with the rationale given by the authors (see also point 2., 5. and 6.). There is no substantiation given for the specific hypotheses.
8. Introduction: The novelty value for the reasons questions as formulated and examined is unclear.
9. Methods, participants. How are they recruited? Are male vs. female and young vs. old recruited in the same way? What was the response (per subgroup)?
10. Methods, Actigraph data. Which cut-points were used for light, moderate and vigorous activity? Were the same cut points used for the young and the older sample? If yes, please provide a rationale for not using age-specific cut points.
11. Methods, BI and QoL. The authors describe the psychometric properties rather confusing. Cronbachs alpha is a measure for reliability, not for validity. Fit-index etc. line 113-114. Of what? A confirmative factor analysis? This would support the validity. Self-classified weight (from very underweight to very overweight). line 117-118, higher scores mean increased satisfaction? This seems incorrect. Line 132, what is intra-cluster correlation? Do you mean intra-class correlation? Please rewrite both sections and improve structure.
12. Results: table 1: Differences in financial status, education and employment can be tested as well.
13. Table 2. Layout can be improved, if assumptions are satisfied, please consider to use analysis of variance, and including age x gender interaction effect.
14. Table 3. It is more interesting to test differences in correlations between genders and age groups. Similar as the research questions, the analyses and presented results do not fit with the rationale of the study.
15. Discussion: the validity of actigraph data for levels of PA and the cut-off points used should be discussed, also in relation to meeting the guidelines.
16. Discussion: Differences in PA between men and women are not discussed but should be discussed, also in relation to literature.
17. Discussion: How do difference in education affect the results? Is PA related to education? How does this affect the age difference?
18. Discussion, line 216-217: More studies are needed to determine differences between men and women. Please, refer to literature regarding this issue, e.g. to meta-analysis of He et al. In general, use more references in the discussion section.
19. Please, stress the novelty of the study, the findings more. What does this study add to scientific knowledge? Why should it be published?
20. The sample size is relatively small, how does this affect the results?
Minor points:
Title: with is redundant, old=older
Line 62-63. Several studies..but only one reference.
Author Response
Response to Reviewer 1 Comments
Point 1: General: English language should be checked carefully for complete manuscript
Response 1: First of all, thank you very much for your comments.
English language has been corrected throughout the text by an expert.
Point 2: Introduction line 48-60. Apparently, the major point is that the relation between PA and BI may depend on age. However, this is not examined in this paper. In the paper only differences in means are tested for age.
Response 2: We separated the associations between PA, BI, and QoL among young and old ages. Please see Table 3 and 4.
Point 3: Introduction line 53-56: However, differences in the associations between PA and BI …not fully understood. What exactly is not understood? What information is missing? How does this affect knowledge? Please provide references as well.
Response 3: Here we had two purposes. First, the relationship between PA and BI in the young and the elderly has not been separately investigated. Also, PA has been mainly measured by self-report questionnaires. Therefore, we sought to objectively measure PA using accelerometer. Finally, we investigated the relationships between objective PA and BI in young people and the elderly separately.
Point 4: Introduction line 56-61: This point can be elaborated more extensively. Differences in mean scores of self-report vs. accelerometry is not the main problem, but the relationship. How is the relationship? How may this affect relationships with BI and QoL?
Response 4: The proposal has been followed. The point has been more elaborated. Please see Introduction.
Point 5: Introduction line 71-72: similar to point 2., age differences in relationships are not tested.
Response 5: Similar to point 2 and 3, the proposal has been followed. Please see Introduction.
Point 6: Introduction: The authors do not give a rationale to examine gender differences.
Response 6: The proposal has been followed. Please see last paragraph of Introduction.
Point 7: Introduction: The research questions do not fit with the rationale given by the authors (see also point 2., 5. and 6.). There is no substantiation given for the specific hypotheses.
Response 7: We have re-written the Introduction based on our purposes and novelties. Please see Introduction.
Point 8: Introduction: The novelty value for the reasons questions as formulated and examined is unclear.
Response 8: We have re-written the Introduction based on our purposes and novelties. Please see Introduction.
Point 9: Methods, participants. How are they recruited? Are male vs. female and young vs. old recruited in the same way? What was the response (per subgroup)?
Response 9: It has been added into the text. Please see Methods, Participants.
Yes, all participants recruited in the same way.
Point 10: Methods, Actigraph data. Which cut-points were used for light, moderate and vigorous activity? Were the same cut points used for the young and the older sample? If yes, please provide a rationale for not using age-specific cut points.
Response 10: It has been added into the text. Please see Methods.
Point 11: Methods, BI and QoL. The authors describe the psychometric properties rather confusing. Cronbachs alpha is a measure for reliability, not for validity. Fit-index etc. line 113-114. Of what? A confirmative factor analysis? This would support the validity. Self-classified weight (from very underweight to very overweight). line 117-118, higher scores mean increased satisfaction? This seems incorrect. Line 132, what is intra-cluster correlation? Do you mean intra-class correlation? Please rewrite both sections and improve structure.
Response 11: It has been corrected in the text. Please see Methods.
Point 12: Results: table 1: Differences in financial status, education and employment can be tested as well.
Response 12: It has been corrected in the text. Please see Results.
Point 13: Table 2. Layout can be improved, if assumptions are satisfied, please consider to use analysis of variance, and including age x gender interaction effect.
Response 13: Thank you for your suggestion. We ran ANOVA for comparisons. Please see Results. Also, we combined the results of age and gender differences in one section (i.e., 3.2 Age and Gender Differences).
Point 14: Table 3. It is more interesting to test differences in correlations between genders and age groups. Similar as the research questions, the analyses and presented results do not fit with the rationale of the study.
Response 14: We calculated and reported bidirectional association between PA with BI and QoL for young and old adults separately. Please see Results.
Point 15: Discussion: the validity of actigraph data for levels of PA and the cut-off points used should be discussed, also in relation to meeting the guidelines.
Response 15: The proposal has been followed. Please see Discussion.
Point 16: Discussion: Differences in PA between men and women are not discussed but should be discussed, also in relation to literature.
Response 16: The proposal has been followed. Please see Discussion.
Point 17: Discussion: How do difference in education affect the results? Is PA related to education? How does this affect the age difference?
Response 17: In this study, we only asked participants for their educational level based on 1: high school or less, and 2: college. So, the measurement of education in this research was very preliminary because we did not intend to investigate the effects of education level on PA. In our opinion, in order to have accurate findings, we should measure the level of education more accurately and with more accurate indicators. Therefore, in our opinion, with the data available in this research, it is not possible to accurately examine the effect of educational level on PA.
Point 18: Discussion, line 216-217: More studies are needed to determine differences between men and women. Please, refer to literature regarding this issue, e.g. to meta-analysis of He et al. In general, use more references in the discussion section.
Response 18: The proposal has been followed. Please see Discussion.
Point 19: Please, stress the novelty of the study, the findings more. What does this study add to scientific knowledge? Why should it be published?
Response 19: The proposal has been followed. Please see Conclusion.
Point 20: The sample size is relatively small; how does this affect the results?
Response 20: We mentioned it as a limitation. Please see Discussion.
Minor points:
Title: with is redundant, old=older
Response: It has been corrected. Please see Introduction. We used ‘older’ in whole text.
Line 62-63. Several studies, but only one reference.
Response: It has been corrected. Please see Introduction.

Reviewer 2 Report
The topic is meaningful. The author analyzes the differences in physical activity, body image and quality of life between young people and old people through the measurement of acceleration. The research in this area is helpful for the suggestions of physical activity of old people. The research method of this paper is clear, the selection of research objects is reasonable, and the research results are basically clear. But there are still some important contents that need to be revised and improved.
1) The abstract doesn't reflect the significance and value of this research, but only mentions "investigating the relationship between physical activity and body image and quality of life of young and old people". So what kind of help can it provide to today's society after the investigation? Please explain in the abstract by the author. Secondly, the conclusion in the abstract does not show the difference in body image and quality of life between young people and old people.
2) "Demographic data" should be placed in the research object of the second part, not in the research result, because it is not the research result, but the basic information of participating in the survey.
3) The title of "3.2 Age Difference" should specify the specific content of age difference, so that readers can know what content is different in terms of age.
4) The title of "3.3 Gender Differences" should specify the specific content of age differences, so that readers can know what is different in terms of gender.
5) The "discussion part" should be discussed separately according to different research results, such as why "3.2 age difference" is caused? Why does it cause "3.3 gender difference"?
6) The conclusion part should increase the research deficiency.
7) The most important content of the full text is not explained clearly: Through the measurement of accelerometer, young people's physical activity is vigorous and their body image is good. Does it mean that their life quality is high? Does less physical activity and poor body image of the elderly mean lower quality of life?
Author Response
Response to Reviewer 2 Comments
Point 1: The abstract doesn't reflect the significance and value of this research, but only mentions "investigating the relationship between physical activity and body image and quality of life of young and old people". So what kind of help can it provide to today's society after the investigation? Please explain in the abstract by the author. Secondly, the conclusion in the abstract does not show the difference in body image and quality of life between young people and old people.
Response 1: First of all, thank you very much for your comments.
It has been corrected in the Abstract.
Point 2: "Demographic data" should be placed in the research object of the second part, not in the research result, because it is not the research result, but the basic information of participating in the survey.
Response 2: We compared demographic data including BMI, financial status, education, and employment between young and older adults. So, we kept demographic data as our findings in the Results.
Point 3: The title of "3.2 Age Difference" should specify the specific content of age difference, so that readers can know what content is different in terms of age.
Response 3: As a suggestion or first reviewer, we used ANOVA for analysing age and gender differences. So, we combined sections 3.2 and 3.3 together. We changed the title of 3.2 as “Young vs. Old and Male vs. Female Differences”
Point 4: The title of "3.3 Gender Differences" should specify the specific content of age differences, so that readers can know what is different in terms of gender.
Response 4: Please refer to Point 3.
Point 5: The "discussion part" should be discussed separately according to different research results, such as why "3.2 age difference" is caused? Why does it cause "3.3 gender difference"?
Response 5: That was the point of first reviewer too. We tried to re-organized our discussion based on our findings. Please see Discussion.
Point 6: The conclusion part should increase the research deficiency.
Response 6: The proposal has been followed. Please see Conclusion.
Point 7: The most important content of the full text is not explained clearly: Through the measurement of accelerometer, young people's physical activity is vigorous and their body image is good. Does it mean that their life quality is high? Does less physical activity and poor body image of the elderly mean lower quality of life?
Response 7: Based on our findings, neither young nor older adults met WHO guidelines. However, young adults were more physically active than older adults. In the Discussion, we attempted to interpret these findings based on young and older adults. As we found significant associations between PA and QoL, it may be considered as a potential factor in lower perceived-QoL in older adults compared with young adults. Please see Discussion.

Reviewer 3 Report
I was interested in an opportunity to review a topic such as “ Associations between Accelerometer-Measured Physical Activity with Body-Image and Quality of Life among Young and Old Adults“. The aim of this study was to examine the association of physical activity with body image and quality of life in young and older adults. The idea of this manuscript is interesting. However, I have some concerns:
Introduction
1. Lines 44-45: it is recommended to supplement and expand information on physical activity (PA) of different age groups based on data published by authors in other countries.
2. Lines 46-47: the data provided by the authors on the prevalence of global overweight and obesity are too old. I suggest analysing more recent data for 2021-2022.
3. Line 51: authors wrote: “...negative emotions such as anxiety and depression“. Anxiety and depression are mental disorders (not negative emotions).
4. Lines 53-56: I would suggest clarifying both “young adult“ and “older adults“ for specific age periods.
5. Information on old people should be added to the introduction section. The clear definition of the existing problems related to the old age should be addressed according to existing scientific literature.
Materials and Methods
6. The authors should add the recruitment process of the study participants.
7. The authors should add the inclusion and exclusion criteria of this study.
8. The authors should add the methods for calculating the sample size.
9. The statistical analysis subsection must be extended. It is recommended that all statistical methods used for data analysis must be described step by step.
10. The category of the “strength of the correlation” should be added in the Statistical Analysis section
11. What type of independent t-test has been used (Welch's t-test or Student's t-test)?
12. The authors compared research variables among young and old adults across genders. However, how did the authors identify gender identity of participants? Which research questions were gender related?
IJERPH encourage the authors to follow the ‘Sex and Gender Equity in Research – SAGER – guidelines’ and to include sex and gender considerations where relevant. Authors should use the terms sex (biological attribute) and gender (shaped by social and cultural circumstances) carefully in order to avoid confusing both terms. Article titles and/or abstracts should indicate clearly what sex(es) the study applies to. Authors should also describe in the background, whether sex and/or gender differences may be expected; report how sex and/or gender were accounted for in the design of the study; provide disaggregated data by sex and/or gender, where appropriate; and discuss respective results. If a sex and/or gender analysis was not conducted, the rationale should be given in the Discussion.
13. Data on measurements related to body mass index (BMI) appear to be missing in the section of “Materials and Methods”. Also, what does “financial status“ (in terms of low, medium, high) mean?
Results, Discussion, and Conclusions
14. The authors analysed only the correlation between the variables. It is therefore necessary to replace the term “the associations” with “the correlations” throughout the entire manuscript text.
15. It is incomprehensible how the authors measured the gender identity of study participants? On what basis did the authors make a gender analysis?
16. As the authors have shown, the study was conducted during the COVID-19 pandemic. It is therefore necessary to clearly describe, both in the introduction section and in the discussion section, the effects of quarantine-related restrictions on the physical activity and quality of life of the target/eligible population. All analysing variables should be evaluated additionally in this context.
17. The generalisability (and implications for the potential readers in foreign countries) of this study should be added.
18. The authors should add the sentences of the limitations of this study.
Author Response
Response to Reviewer 3 Comments
Point 1: Lines 44-45: it is recommended to supplement and expand information on physical activity (PA) of different age groups based on data published by authors in other countries.
Response 1: First of all, thank you very much for your comments. Proposal has been followed. Please see Introduction.
Point 2: Lines 46-47: the data provided by the authors on the prevalence of global overweight and obesity are too old. I suggest analysing more recent data for 2021-2022.
Response 2: Proposal has been followed. Please see Introduction.
Point 3: Line 51: authors wrote: “...negative emotions such as anxiety and depression“. Anxiety and depression are mental disorders (not negative emotions).
Response 3: It has been corrected in the text. Please see Introduction.
Point 4: Lines 53-56: I would suggest clarifying both “young adult“ and “older adults“ for specific age periods.
Response 4: Proposal has been followed. Please see Introduction.
Point 5: Information on old people should be added to the introduction section. The clear definition of the existing problems related to the old age should be addressed according to existing scientific literature.
Response 5: Proposal has been followed. Please see Introduction.
Point 6: The authors should add the recruitment process of the study participants.
Response 6: First reviewer has also commented this. We have added the recruitment of the participants in the Method. However, it is with red colour. Please see Method.
Point 7: The authors should add the inclusion and exclusion criteria of this study.
Response 7: Proposal has been followed. Please see Method.
Point 8: The authors should add the methods for calculating the sample size.
Response 8: Proposal has been followed. Please see Method.
Point 9: The statistical analysis subsection must be extended. It is recommended that all statistical methods used for data analysis must be described step by step.
Response 9: Proposal has been followed. Please see Method.
Point 10: The category of the “strength of the correlation” should be added in the Statistical Analysis section
Response 10: Proposal has been followed. Please see Method.
Point 11: What type of independent t-test has been used (Welch's t-test or Student's t-test)?
Response 11: As first reviewer suggested, we used ANOVA for data analysis. Please see Results.
Point 12: The authors compared research variables among young and old adults across genders. However, how did the authors identify gender identity of participants? Which research questions were gender related?
IJERPH encourage the authors to follow the ‘Sex and Gender Equity in Research – SAGER – guidelines’ and to include sex and gender considerations where relevant. Authors should use the terms sex (biological attribute) and gender (shaped by social and cultural circumstances) carefully in order to avoid confusing both terms. Article titles and/or abstracts should indicate clearly what sex(es) the study applies to. Authors should also describe in the background, whether sex and/or gender differences may be expected; report how sex and/or gender were accounted for in the design of the study; provide disaggregated data by sex and/or gender, where appropriate; and discuss respective results. If a sex and/or gender analysis was not conducted, the rationale should be given in the Discussion.
Response 12: As first reviewer commented, we corrected our research aims and hypothesis. Please see Introduction. As well, using ANOVA, we measured gender differences, too.
Point 13: Data on measurements related to body mass index (BMI) appear to be missing in the section of “Materials and Methods”. Also, what does “financial status“ (in terms of low, medium, high) mean?
Response 13: Proposal has been followed. Please see Method. We added BMI calculation in section 2.2.4.
Point 14: The authors analysed only the correlation between the variables. It is therefore necessary to replace the term “the associations” with “the correlations” throughout the entire manuscript text.
Response 14: It has been corrected in whole text.
Point 15: It is incomprehensible how the authors measured the gender identity of study participants? On what basis did the authors make a gender analysis?
Response 15: We used a 2 (Age: young vs. old) × 2 (Gender: male vs. female) analysis of variance (ANOVA) to conduct gender analysis in PA pattern, BI, and QoL between young and older adults across genders. Please see Data Analysis.
Point 16: As the authors have shown, the study was conducted during the COVID-19 pandemic. It is therefore necessary to clearly describe, both in the introduction section and in the discussion section, the effects of quarantine-related restrictions on the physical activity and quality of life of the target/eligible population. All analysing variables should be evaluated additionally in this context.
Response 16: This study was not conducted during quarantine due to the COVID-19 pandemic. We conducted this study during the post-quarantine era, when all restrictions caused by the COVID-19 pandemic had been lifted.
Point 17: The generalisability (and implications for the potential readers in foreign countries) of this study should be added.
Response 17: It has been added in the Conclusions.
Point 18: The authors should add the sentences of the limitations of this study.
Response 18: We have a paragraph about the strength and limitations of this study. Please see last paragraph of the Discussion.

Round 2
Reviewer 1 Report
See also file (bullet numbers seem missing now)
Although the authors improved the paper, I still feel objectives and results/data do not match sufficiently and the authors do not convince regarding the need and novelty of the study. Furthermore, sample size is (too) low for this kind of cross-sectional study.
I have the following more specific concerns.
1. The primary purpose in the revised version is now to investigate differences in PA between young and old adults (line 57-58). According to the authors this is not done before. This is incorrect. Just one example Varma et al. (2017) in Preventive Medicine used accelerometer data in over 12.000 male and female participants of different age groups. And this is just an example.
2. Overall, the introduction is messy and does not lead to clearly formulated hypotheses. To illustrate the authors appeared to have the following aims/objectives/purposes:
a. Investigate difference in PA between young and old (line 57-58)
b. To examine the relationship between PA and BI stratified by age-group (line 78-79) .
c. To examine the associations between PA and BI with QoL among young and old adults (line 91-92)
d. To examine gender differences in PA, BI and QoL? (line 93-94) (but without proper review of literature, what is novel here?)
e. To investigate gender differences in PA, BI and QoL in relation to age (line 95-95) (so gender x age interaction, but again without rationale).
The authors only formulated hypotheses in relation to aim a. and d.) (line 99-101). The other hypotheses are not introduced (BI in relation to age, QoL in relation to age and even correlations of PA with BI/QoL). So, what looks novel (does the relation between PA and BI depend on age, with a hypothesis about how) (aim b.) is ignored here.
To examine if the relation between PA and QoL depends on age is not mentioned at all, although the results are provided.
3. Abstract: info regarding aim b. (above) is missing. Did the relation between PA-BI depend on age? And how?
4. Table 2: although the interactions appeared important for the aims, they are ignored in text results and discussion sections.
5. Table 3 and Table 4. Since age differences in associations seem important it is difficults for interpretation if the info is in different tables. I suggest to combine them in one table. Maybe it helps to transpose the variables.
6. Discussion line 238-239. Apparently, aim b (above) is the most important question. But the emphasis in the discussion is about age and gender differences in PA, BI and QoL. And much less on the associations. See also point 3.
7. Conclusions: the sample size is too small and the recruitment strategy does not guarantee a representative sample. This makes generalization and recommendations about PA inappropriate.
Minor comments
What is meant by “associations between PA and BI with QoL” (e.g., line 92)? PA-QoL and BI-QoL? Please rephrase to make clear.
Line 145: Cronbachs alpha total score should ve Cronbachs alpha total scale.
Section 2.3. Pearson correlations: stratified by age group is missing.
Cut-off Vigorous PA is missing (line 128)

Author Response
Manuscript ID: ijerph-1863903
Response to Reviewer 1
Point 1: The primary purpose in the revised version is now to investigate differences in PA between young and old adults (line 57-58). According to the authors this is not done before. This is incorrect. Just one example Varma et al. (2017) in Preventive Medicine used accelerometer data in over 12.000 male and female participants of different age groups. And this is just an example.
Response 1: The proposal has been followed. Please see Introduction.
Point 2: The authors only formulated hypotheses in relation to aim a. and d.) (line 99-101). The other hypotheses are not introduced (BI in relation to age, QoL in relation to age and even correlations of PA with BI/QoL). So, what looks novel (does the relation between PA and BI depend on age, with a hypothesis about how) (aim b.) is ignored here.
Response 2: The hypotheses of the study were edited according to the aims of the study. Please see Introduction. It should be noted that we removed gender analysis from the study.
Point 3: Abstract: info regarding aim b. (above) is missing. Did the relation between PA-BI depend on age? And how?.
Response 3: The proposal has been followed. Please see Abstract.
Point 4: Table 2: although the interactions appeared important for the aims, they are ignored in text results and discussion sections
Response 4: We removed gender analysis from the study and accordingly we used independent t tests for analysing age differences.
Point 5: Table 3 and Table 4. Since age differences in associations seem important it is difficults for interpretation if the info is in different tables. I suggest to combine them in one table. Maybe it helps to transpose the variables.
Response 5: Due to large data in these tables, we kept the data separately in two tables.
Point 6: Discussion line 238-239. Apparently, aim b (above) is the most important question. But the emphasis in the discussion is about age and gender differences in PA, BI and QoL. And much less on the associations. See also point 3.
Response 6: We removed gender analysis from the study and our discussion is mainly on age differences.
Point 7: Conclusions: the sample size is too small and the recruitment strategy does not guarantee a representative sample. This makes generalization and recommendations about PA inappropriate.
Response 7: We reported our relatively small sample size as a limitation.
Minor points:
What is meant by “associations between PA and BI with QoL” (e.g., line 92)? PA-QoL and BI-QoL? Please rephrase to make clear
Response: The proposal has been followed. Please see Introduction.
Cronbachs alpha total score should be Cronbachs alpha total scale.
Response: The proposal has been followed. Please see Method.
Pearson correlations: stratified by age group is missing.
Response: Correlations between research variables are presented separately for young and older adults.
Cut-off Vigorous PA is missing.
Response: The proposal has been followed. Please see Method.

Reviewer 2 Report
The opinions put forward by experts have been revised, basically meeting the publishing requirements.
Author Response
ijerph-1863903
Dear respected reviewer,
Thanks for considering our manuscript for publishing.
Thanks for your time and cooperation.
Best regards.
Reviewer 3 Report
The authors should add the methods for calculating the representative sample size.
Line 176: BMI is not demographic data....
„Associations“ must be changed to „correlations“
The authors compared research variables among young and old adults across genders. However, how did the authors identify gender identity of participants? Which research questions were gender related?
The term „gender” must be changed to “sex”. Unless the authors included trangenders, etc. in the study.
It is incomprehensible how the authors measured the gender identity of study participants? On what basis did the authors make a gender analysis?
I doubt that the comparison of data by gender makes sense, since the author's comparison groups are age groups. In addition, why was the ANOVA method selected for 2 x 2 tables?
The authors stated that “this study is limited in the sense that our sample size was relatively small and our findings are not necessarily generalizable to all young and older adults“.
If the results of the study cannot be generalised, the study itself is not scientifically relevant.
Author Response
Manuscript ID: ijerph-1863903
Response to Reviewer 3
Point 1: The authors should add the methods for calculating the representative sample size.
Response 1: The sample size calculation was based on recommendations for cross-sectional studies using accelerometers for measuring PA.
Point 2: Line 176: BMI is not demographic data.....
Response 2: We removed BMI from the demographic data.
Point 3: „Associations“ must be changed to „correlations“.
Response 3: Due to the fact that we use the word “association” in the title, we kept this phrase throughout the manuscript.
Point 4: The authors compared research variables among young and old adults across genders. However, how did the authors identify gender identity of participants? Which research questions were gender related?
The term „gender” must be changed to “sex”. Unless the authors included trangenders, etc. in the study.
It is incomprehensible how the authors measured the gender identity of study participants? On what basis did the authors make a gender analysis?
I doubt that the comparison of data by gender makes sense, since the author's comparison groups are age groups. In addition, why was the ANOVA method selected for 2 x 2 tables?
Response 4: We removed gender analysis from the study and accordingly we used independent t tests for analysing age differences.
Point 5: The authors stated that “this study is limited in the sense that our sample size was relatively small and our findings are not necessarily generalizable to all young and older adults“.
If the results of the study cannot be generalised, the study itself is not scientifically relevant.
Response 5: We reported our relatively small sample size as a limitation.
